# QUBE: Enhancing Automatic Heuristic Design via Quality-Uncertainty Balanced Evolution

## Abstract

Solving NP-hard problems traditionally relies on heuristics, yet manually designing effective heuristics for complex problems remains a significant challenge. While recent advancements like FunSearch have shown that large language models (LLMs) can be integrated into evolutionary algorithms (EAs) for heuristic design, their potential is hindered by limitations in balancing exploitation and exploration. We introduce Quality-Uncertainty Balanced Evolution (QUBE), a novel approach that enhances LLM+EA methods by redefining the priority criterion within the FunSearch framework. QUBE employs the Quality-Uncertainty Trade-off Criterion (QUTC), based on our proposed Uncertainty-Inclusive Quality metric, to evaluate and guide the evolutionary process. Through extensive experiments on challenging NP-complete problems, QUBE demonstrates significant performance improvements over FunSearch and baseline methods. Our code will be made public upon acceptance.

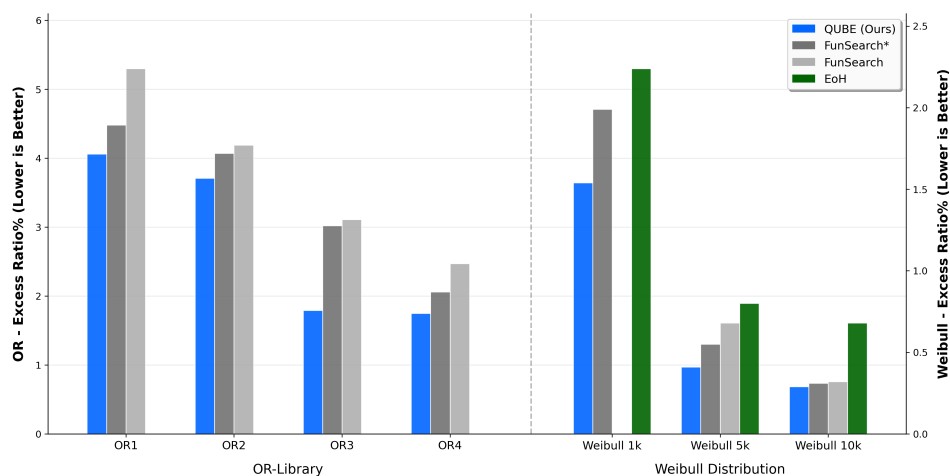

Figure 1: Experiment results on online bin packing, the performance is evaluated with "Excess Ratio". Our method can steadily find better heuristics than all baselines. Note: FunSearch's result on Weibull1k, EoH's result on OR Library is unavailable. FunSearch* is our reproduction of Fun-Search, details in Section 4.3. Refer to Appendix E for results on Cap Set and TSP.

## 1 Introduction

Many mathematical science problems are NP-complete, making them extremely challenging to solve but relatively easy to evaluate (Romera-Paredes et al., 2024). Evolutionary Algorithms (EAs) are widely used to optimize heuristics for such problems (Liu et al., 2023; Mei et al., 2023). Recently, large language models (LLMs) have demonstrated remarkable capabilities in code generation (Austin et al., 2021; Chen et al., 2021; Li et al., 2023), opening up new avenues for hyper-heuristic algorithms. These methods, termed "LLM+EA" methods, leverage LLMs as variation operators within EAs, achieving promising results across diverse domains (Chen et al., 2024; Zheng et al., 2023; Nasir et al., 2024; Wang et al., 2024). A notable example is FunSearch (Romera-Paredes

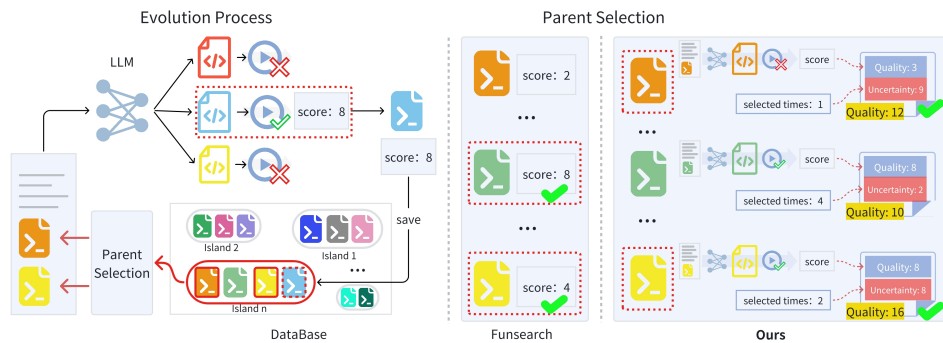

Figure 2: **Overview of the QUBE. Left**: The overall evolutionary pipeline of QUBE, which builds upon the structure of FunSearch. **Right**: Comparison of parent selection strategies. While Fun-Search selects parents based solely on sample scores, QUBE selects parents based on Uncertainty-Inclusive Quality (UIQ) we proposed.

et al., 2024), which discovers high-quality heuristics through approximately 2.5 million evolutionary steps in a multi-population EA framework.

Theoretically, to optimize heuristics in a "function space", a method should excel in two key aspects: exploitation (deepening search in promising regions) and exploration (broadening search in unknown regions). However, achieving this balance long remains an open challenge (Weng, 2020; Sutton et al.). Through analysis, we observe that the priority criterion behind FunSearch's evolution process hinders it from exploiting upon current status and performing useful exploration within the function space, resulting its struggle.

To address the limitations of FunSearch in balancing exploration and exploitation, we propose Quality-Uncertainty Balanced Evolution (QUBE): a novel framework that enhances heuristic evolution by redefining the priority criterion guiding the evolutionary process. At the core of QUBE is the Quality-Uncertainty Trade-off Criterion (QUTC), which builds on our proposed Uncertainty-Inclusive Quality (UIQ) to assess samples. QUBE is experimented on both standard combinatorial optimization problems and more challenging problems such as Cap Set. As illustrated in Figure 1, QUBE demonstrates significant improvements over baseline methods[1].

We summarize our contributions as follows:

1. We identify a key limitation in FunSearch: its priority criterion fails to effectively balance exploitation and exploration, thereby constraining its performance in heuristic evolution.
2. We propose QUBE, an LLM+EA method that employs Quality-Uncertainty Trade-off Criterion (QUTC) enabling balanced exploration & exploitation throughout the evolutionary process.
3. Experimental results across multiple NP-complete problems demonstrate significant improvements: reduction in excess bin usage for online bin packing (OBP), enhanced solution quality for traveling salesman problem (TSP), and larger cap set discoveries.

## 2 THOROUGH EXAMINING EXPLORATION AND EXPLOITATION IN FUNSEARCH

In this section, we first provide an overview of FunSearch and elaborate on two important details: parent selection during each evolution step, and island reset that periodically takes place. A priority criterion that affects these core details is identified. Then, we define exploitation and exploration in FunSearch and analyze how the priority criterion affects the balance between exploitation and exploration. Finally, we empirically show FunSearch's deficiency in both exploitation and exploration.

---

[1]We only show results for online bin packing here, please refer to Appendix E for more results. See Section 4 for experimental details.

## 2.1 OVERVIEW OF FUNSEARCH

FunSearch is an LLM+EA method designed to evolve heuristics of complex combinatorial problems, represented as Python functions. It employs an LLM as a variation operator within an EA framework that utilizes multiple populations, or "islands." An overview of FunSearch's evolutionary process is illustrated in the left part of Figure 2.

At each iteration, a randomly selected island undergoes evolution. Two parent samples are chosen, and the LLM generates new candidates using these parents as few-shot examples. The generated samples are evaluated for performance, and only those that execute successfullywithout exceptions or timeoutsare retained. Periodically, underperforming islands are reset by replacing all their samples with the best-performing sample from a high-performing island. Specifically, half of the lowest-performing islands are reset in this manner.

A central component of FunSearch is its priority criterion, which governs both parent selection and island resets. FunSearch's criterion is based on sample scores: the probability of a sample being selected as a parent is proportional to the exponential of its score. An island is reset if its highest-scoring sample performs worse than at least half of the other islands. This mechanism prioritizes high-performing samples while preserving some diversity across populations.

## 2.2 EXPLORATION AND EXPLOITATION IN FUNSEARCH

The primary objective of FunSearch is to identify high-performance heuristics through iterative sampling. Achieving this requires effective **exploitation** of known regions in the function space: generating samples that incrementally improve upon prior performance. However, limiting the search to familiar areas risks overlooking heuristics with superior potential. To mitigate this, FunSearch must also engage in **exploration**, generating diverse samples from less-explored regions that may initially appear unpromising. Balancing these two strategies, exploitation and exploration, is essential for robust heuristic discovery.

The priority criterion plays a pivotal role in managing this balance. At each evolutionary step, it guides the selection of parent samples used to prompt the LLM for new candidates. To maximize exploitation, the criterion should prioritize parents with a high likelihood of producing strong offspring. Conversely, to encourage exploration, it should also allow for the selection of parents with uncertain performance, enabling the algorithm to probe novel regions of the function space.

Methods like FunSearch incorporate island reset mechanisms. The priority criterion also determines which islands to reset. Ideally, islands that have thoroughly exploited their regions but yield consistently low scores should be reset to refocus on promising areas. Yet islands with low performance but incomplete exploration may be preserved to allow continued search in underexplored spaces.

Ultimately, the priority criterion must strike a careful balance between exploitation and exploration, as overemphasis on either strategy can compromise the effectiveness of the other.

## 2.3 QUANTITATIVE ASSESSMENT OF EXPLORATION AND EXPLOITATION

To analyze the search dynamics of Funsearch, we introduce two evaluation metrics: Recent Best Score (RBS) and Recent Proportion of Change (RPC). These metrics are designed to approximate the algorithm's exploitation and exploration behavior, respectively, over a short time window (window size is set to $K = 500$ in practice).

**Recent Best Score (RBS)** measures the highest score among the $K$ most recent samples. It reflects the algorithm's capacity to consistently generate high-performing heuristics within its current search region. While exploration may occasionally yield high scores, sustained improvements within a short window are more indicative of effective exploitation.

**Recent Proportion of Change (RPC)** quantifies the token-level edit distance between each of the recent $K$ samples and its nearest parent, normalized by sample length. This serves as a tractable proxy for exploration, since full population-level diversity tracking is computationally infeasible at our scale[2]. Thus we focus on structural novelty relative to parent samples. A higher RPC indicates broader search behavior, while lower values suggest conservative sampling.

---

[2]Calculating pairwise edit distance among full population has complexity of $O(N^2)$, and in practice would require days to fully analyze one experiment: $0.1\text{ms} \times 80000^2/2 = 3.2e5\text{s} \approx 88.9\text{hour}$

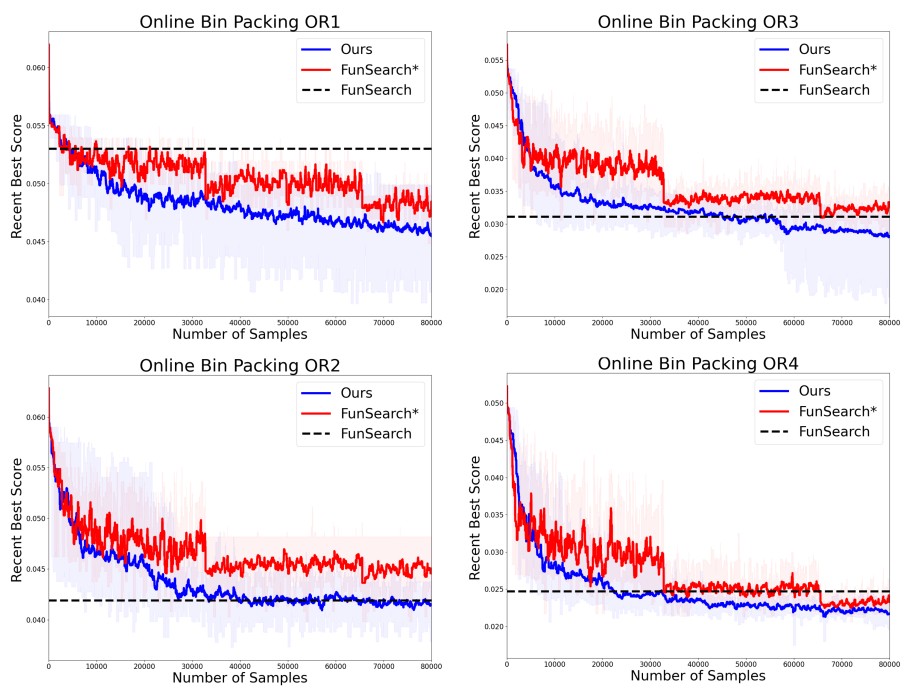

Figure 3: The "Recent Best Score" of FunSearch exhibits plateaus in later stages, indicating challenges in effectively exploiting known regions. In contrast, our method consistently generates higher-scoring samples, demonstrating superior exploitation capabilities.

As shown in Figure 3, FunSearch's RBS exhibits sporadic jumps betwee long plateaus, implying less consistent exploitation. However, our method QUBE shows a steady progress in RBS, suggesting stable and effective exploitation. This discrepancy arises from FunSearch's reliance on sample score as its priority criterion, which does not necessarily correlate with the score of child samples.

Figure 4 further shows that FunSearch maintains a consistently low RPC, implying limited exploratory reach. Although FunSearch employs techniques such as multi-population evolution and temperature-based sampling, its exploration strategy lacks discrimination. An effective exploration strategy should prioritize regions with higher potential for high-scoring samples while avoiding excessive search in low-potential areas. Ideally, exploration should dominate the early stages when the function space is largely unexplored, and gradually shift toward exploitation as promising regions emerge. QUBE exemplifies this adaptive behavior: it starts high in RPC, indicating aggressive exploration, and gradually reduces it over time, demonstrating balanced exploitation and exploration.

## 3 QUALITY-UNCERTAINTY BALANCED EVOLUTION OF HEURISTICS

To balance exploration and exploitation in hubristic evolution, we propose Quality-Uncertainty Balanced Evolution (QUBE). We begin by outlining the overall framework of QUBE. We then introduce the Quality-Uncertainty Trade-off Criterion (QUTC), a priority mechanism grounded in our proposed Uncertainty-Inclusive Quality (UIQ) for sample evaluation. Finally, we demonstrate how QUBE incorporates QUTC into key components of the evolutionary process, including parent selection and island reset.

### 3.1 OVERALL FRAMEWORK

At a macro level, the overall structure of our method (Figure 2's left) aligns with that of FunSearch: an LLM+EA method with multi-populations. QUBE evolves a Python function that serves as a heuristic within a search algorithm. The performance of each function sample $c$ is deterministically evaluated by executing the search algorithm on a fixed set of test instances, yielding a score $s(c)$. All samples are stored in a database $\mathbb{D}$, which consists of $n \geq 1$ islands. Each island $\mathbb{I}$ maintains

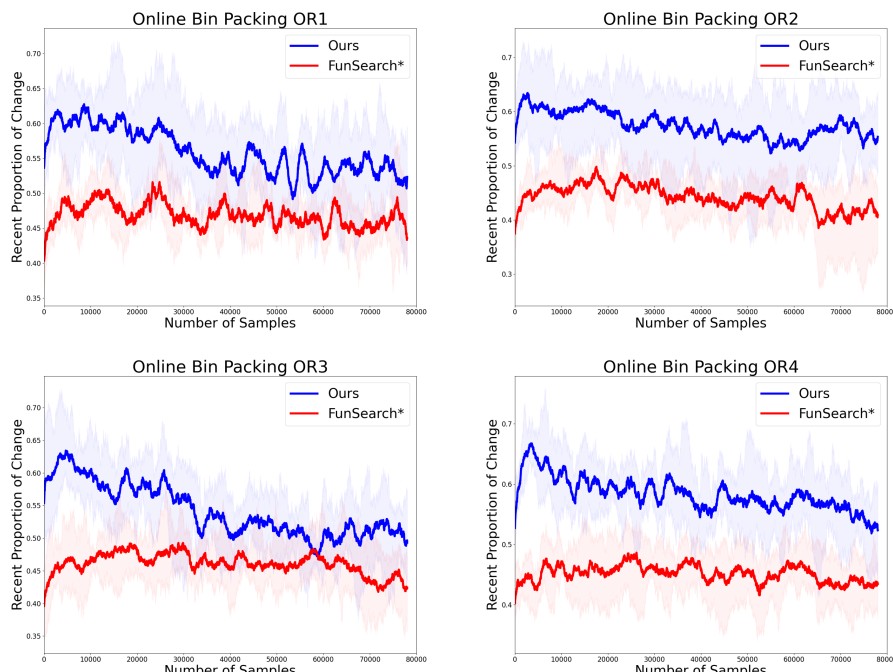

Figure 4: FunSearch has a consistently low "Recent Proportion of Change", reflecting its limited overall exploration of the function space. In contrast, our method demonstrates both a broader scope and a more intelligent exploration strategy, enabling more effective discovery of promising regions.

an independent population, with no inter-island communication except during resets. Within each island, samples are further grouped into clusters $\mathbb{C}$, where all members produce identical outputs across test instances and thus share the same score $s(\mathbb{C})$.

At each timestep, an island $\mathbb{I}$ is randomly selected for sample generation. Two parent samples are chosen from $\mathbb{I}$ using our priority criterion QUTC, and passed as few-shot examples to the LLM to generate new samples. After evaluation, the newly generated samples are stored back on the same island $\mathbb{I}$. Periodically, after every $T_{reset}$ sample generation, our method identifies and resets half of the underperforming islands according to QUTC. This reset strategy helps maintain a dynamic balance between exploration and exploitation by refreshing stagnant regions of the search space.

## 3.2 QUALITY-UNCERTAINTY TRADE-OFF CRITERION

To effectively balance exploitation and exploration, our priority criterion QUTC must identify samples that offer evolutionary advantages, specifically those likely to produce high scores in newly generated samples, while also considering less-explored regions of the search space, represented by samples that have been visited less frequently. In practice, we observed significant similarity among samples within the same cluster. Thus, QUTC prioritizes clusters as a whole rather than individual samples, ensuring a more efficient and scalable approach to guiding the evolutionary process.

We first introduce **Uncertainty-Inclusive Quality (UIQ)**, the metric used by QUTC to assess samples. We define the **quality** of each cluster as the mean score of all child samples generated before the current timestep, using samples from this cluster as parents. Formally, this is expressed as:

$$Q_t(\mathbb{C}) = \frac{1}{\sum\limits_{c \in \mathbb{C}} |\mathbb{P}_{c,t}|} \sum_{c \in \mathbb{C}} \sum_{a \in \mathbb{P}_{c,t}} s(a) \tag{1}$$

where $\mathbb{P}_{c,t}$ denotes the collection of all samples generated with $c$ as a parent before timestep $t$. Unlike existing LLM+EA methods that statically assess a sample's quality using its own score, $Q_t(\mathbb{C})$ dynamically estimates the expected performance of child samples produced by samples in $\mathbb{C}$. This enables the identification of clusters with evolutionary advantages.

Inspired by Upper Confidence Bound (UCB) (Lai & Robbins, 1985; Auer, 2002), we incorporate **uncertainty** based on visiting times of each cluster into its **quality** $Q_t(\mathbb{C})$, resulting in **UIQ**, $\tilde{Q}_t(\mathbb{C})$. Specifically, let $N_t(\mathbb{C})$ be the number of times samples in cluster $\mathbb{C}$ are used as parents before timestep $t$, and $k$ be a hyperparameter. We define UIQ as:

$$\tilde{Q}_t(\mathbb{C}) = Q_t(\mathbb{C}) + k\sqrt{\frac{\ln t}{N_t(\mathbb{C})}} \tag{2}$$

By construction, UIQ integrates both the clusters' estimated evolutionary quality and the uncertainty associated with this estimate. QUTC ranks and prioritizes clusters based on their UIQ values, thereby automatically balances exploitation and exploration, which was lacking in FunSearch.

### 3.3 Quality-Uncertainty Balanced Evolution

Our method QUBE incorporates QUTC into the parent selection at each evolution step and the evaluation of islands at each island reset procedure.

As illustrated in the right part of Figure 2. After an island $\mathbb{I}$ is selected to evolve new samples at each timestep $t$, we identify 2 clusters in $\mathbb{I}$ with the highest UIQ according to QUTC. We select one sample per cluster to serve as parents for this step. We bias the choice toward shorter programs in the cluster (via the length-based weight) to favor simpler heuristics when quality is similar. Specifically, let $l_c$ be the length of sample $c$ measured by the number of characters, the probability of chosen $c$ within a cluster is proportionate to $\exp(\frac{l_c}{T_{prog}})$, where $T_{prog} > 0$ is a hyperparameter.

At each island reset interval, we assess island quality using the cluster with the highest UIQ. Islands whose top UIQ falls below the global median are selected for reset. Each reset island is reinitialized by sampling from the best-performing cluster of a randomly chosen surviving island, ensuring a promising foundation for continued evolution. This differs from FunSearch, which resets based on raw performance rank; our island reset avoids discarding islands that still have high potential (uncertainty) despite lower scores currently.

## 4 Experiments

### 4.1 Implementation Details

We implement an asynchronous system on a single server with 8 NVIDIA A100 GPUs. LLM inference service is set up locally using SGLang (Zheng et al., 2024). This enables efficient parallel sampling and decouples LLM inference from the rest of the system. We use OpenCoder-8B-Instruct (Huang et al., 2024) for all experiments, and include ablation studies with Deepseek-coder-6.7B (Guo et al., 2024). We provide our prompt for LLM in Appendix I.

The remaining components operate in parallel via multiprocessing. Samplers retrieve parent samples from a shared database and submit requests to the LLM backend. Generated samples are evaluated and stored back into the database. For TSP, we use a simplified configuration with a single island and no reset, as satisfactory results can be achieved with fewer samples. More hyperparameter settings are provided in Table 4 of the appendix.

### 4.2 Experiment problems

We assessed the performance of our method on three NP-complete problems:

**Online Bin Packing (OBP)**: OBP aims to accommodate each one of a set of items immediately into the least number of fixed-sized bins. We conduct experiments on the OR-Library (Beasley, 1990) which comprises four OBP datasets (OR1 to OR4), as well as generated instances from the Weibull distribution. Following FunSearch (Romera-Paredes et al., 2024), heuristics are evolved within a local search framework. Performance of OBP is measured using the "excess ratio", defined as the fraction of bins used beyond the L2 lower bound (Martello & Toth, 1990) of the optimal solution.

**Cap Set (CS)**: The goal of the Cap Set problem is to find the largest subset of vectors in $\mathbb{Z}_3^n$ such that no three vectors sum to zero. As in FunSearch, we evolve a priority function that ranks vectors

| | OBP ($\downarrow$) | | | | | | | CS ($\uparrow$) | TSP ($\downarrow$) | | |
| | OR1 | OR2 | OR3 | OR4 | W-1k | W-5k | W-10k | n=8 | TSP20 | TSP50 | TSP100 |
|---|---|---|---|---|---|---|---|---|---|---|---|
| FunSearch* | 4.48% | 4.07% | 3.02% | 2.06% | 1.99% | 0.55% | 0.31% | 464 | 0.000% | 0.000% | 0.029% |
| FunSearch | 5.30% | 4.19% | 3.11% | 2.47% | - | 0.68% | 0.32% | **512** | - | - | - |
| EoH | - | - | - | - | 2.24% | 0.80% | 0.61% | - | 0.000% | 0.000% | 0.025% |
| QUBE | **4.06%** | **3.73%** | **1.79%** | **1.75%** | **1.54%** | **0.41%** | **0.29%** | 480 | 0.000% | 0.000% | **0.023%** |

Table 1: Main experiment results on each task. The best result for each setting is in **bold**. Our method outperforms "FunSearch*", our reproduction of FunSearch on all problems, and is better than FunSearch on online bin packing as well as EoH on TSP.

to guide greedy construction. Experiments are conducted with $n = 8$, and performance is evaluated by the size of the largest cap set found.

**Traveling Salesman Problem (TSP)**: TSP aims at finding the shortest routes that visit all given locations once and return to the starting point. We test QUBE on three settings: TSP20, TSP50, and TSP100, following prior work (Kool et al., 2018; Liu et al., 2024). Our method evolves the objective function used in the perturbation phase of a guided local search algorithm (Voudouris et al., 2010). Performance is assessed using the "excess ratio", defined as the relative distance to the optimal solution computed by Concorde[3].

Each experiment is run 10 times, and the best result is reported unless otherwise specified. Ablation studies include average performance and standard deviation to assess robustness. Additional details on data generation and task-specific implementation are provided in Appendix D.1 and G.

### 4.3 BASELINES

We compared our method with extensive baselines, including: (1) **FunSearch**: For OBP and CS, we report the performance results directly from FunSearch (Romera-Paredes et al., 2024). Since our hardware and LLM setup differ from those used in the original paper, we also re-implement FunSearch on our infrastructure following its published methodology. This reproduced version is denoted as **FunSearch***. (2) **EoH**: For OBP and TSP, we include comparisons with results reported by EoH (Liu et al., 2024; Zhang et al., 2024).

### 4.4 MAIN RESULTS

Table 1 summarizes the performance of the best heuristics obtained by each method. By reimagining the parent selection and island reset with QUTC, QUBE consistently outperforms FunSearch and other baselines across all benchmarks.This demonstrates that our seemingly simple modifications yield substantial improvements in heuristic quality.

On **OBP**, QUBE reduces the excess bin usage by **9.36% $\sim$ 41.73%** compared to FunSearch*, and **10.98% $\sim$ 42.44%** relative to the original FunSearch results on both OR datasets and Weibull-generated instances. On **TSP**, although all methods approach the optimal solution, QUBE still achieves superior performance, with the "excess ratio" **20.69% smaller** than "FunSearch*" and **8.00% smaller** than EoH. These results highlight QUBE's ability to generate high-quality heuristics even in domains with strong existing baselines.

In the **Cap Set** problem, QUBE finds a set that is **16 elements larger** than that found by FunSearch* for $n = 8$. While we do not surpass the best result reported in FunSearch, we argue that reproducing their result is prohibitively expensive due to the extreme computational cost of a full cap set experiment [4]. Despite limited trials, QUBE still discovers larger sets than FunSearch* under same settings, further validating its effectiveness.

### 4.5 DISCUSSION

To mitigate the influence of randomness on best-run performance, we conduct experiments to verify that the observed gains stem from our method's effectiveness in both exploitation and exploration.

---

[3]https://www.math.uwaterloo.ca/tsp/concorde.html

[4]Running a cap set experiment requires generating and evaluating 2.5 million programs, which takes over 3 days on our GPU server. As stated in (Romera-Paredes et al., 2024): among 140 experiments they ran on cap set problem with n=8, less than 5% yield cap set larger than 480.

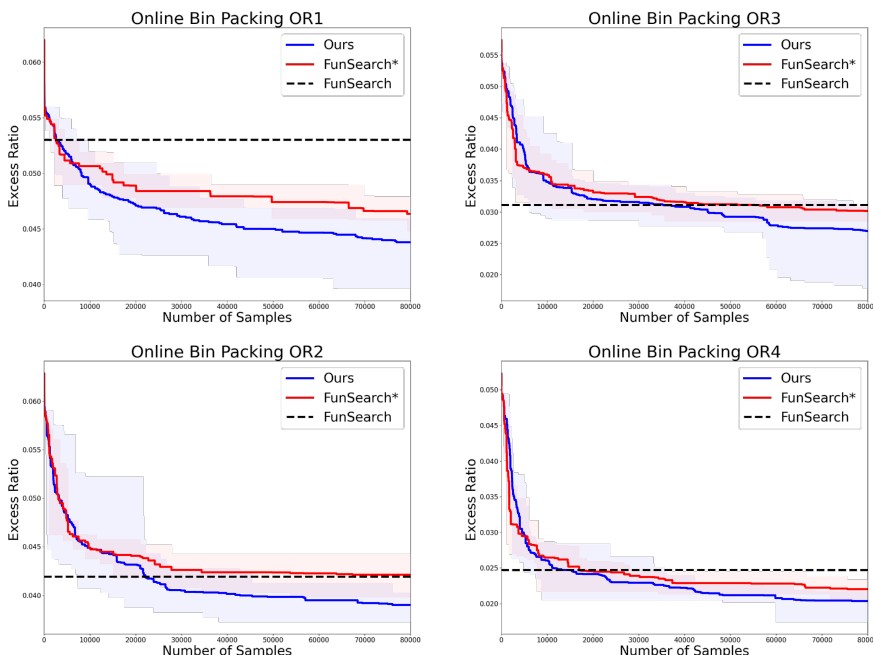

Figure 5: Performance progress on online bin packing. The solid line shows the average score among 10 experiments at each timestep. The shadow shows the range of best and worst experiments. FunSearch is shown in dash line since only a final score is available.

| | Criterion for Parent Selection | Criterion for Island Reset | Best | $\text{Avg}_{\text{std}}$ |
|---|---|---|---|---|
| QUBE (Ours) | $\tilde{Q}_p(\mathbb{C}, t)$ | $\tilde{Q}_p(\mathbb{C}, t)$ | **1.79%** | **$2.76\%_{0.0019}$** |
| Parent Selection Only | $\tilde{Q}_p(\mathbb{C}, t)$ | $s(\mathbb{C})$ | 2.65% | $2.89\%_{0.0018}$ |
| Quality Only | $\tilde{Q}_t(\mathbb{C})$ | $s(\mathbb{C})$ | 2.74% | $2.98\%_{0.0012}$ |
| FunSearch* | $s(\mathbb{C})$ | $s(\mathbb{C})$ | 3.02% | $3.07\%_{0.0008}$ |

Table 2: Ablation of our method on online bin packing OR3. Best stands for best among 10 runs. "$\text{Avg}_{\text{std}}$" stands for the average score, with standard deviation shown as the suffix.

We use the OBP task from the OR-Library as the benchmark, comparing our method against Fun-Search*, our reproduced version of FunSearch.

As shown in Figure 3, QUBE consistently achieves higher RBS than FunSearch*. While Fun-Search* exhibits early progress followed by prolonged plateaus, QUBE continues to discover improved heuristics throughout the evolutionary process, suggesting that QUBE maintains stronger exploitation capabilities. We attribute this to our priority criterion QUTC and its UIQ-based sample evaluation that better aligns with the goal of exploitation.

Figure 4 presents the RPC for both methods. FunSearch* maintains a low and steady RPC, indicating undirected and random exploration. In contrast, QUBE begins with aggressive exploration and gradually reduces, adaptively shifting towards exploitation and focused refinement within explored promising areas. This adaptive behavior reflects a more strategic exploration pattern, consistent with our analysis in Section 2.3.

Further evidence is provided in Figure 5, which tracks the performance trajectory over time. QUBE outperforms both FunSearch and FunSearch* from early stages, while also offering a higher improvement rate especially in later phases. Together, these results demonstrate that QUBE achieves a more effective balance between exploration and exploitation, leading to consistently superior and robust heuristic performance.

## 4.6 ABLATION STUDY

To better understand the contributions of individual components in QUBE, we conduct an ablation study on the OR3 dataset of the Online Bin Packing (OBP) task. All variants share the same implementation setup as described in Section 4.1, unless otherwise specified.

| LLM | Method | Best Run | $\text{Avg}_{\text{std}}$ |
|---|---|---|---|
| OpenCoder-8B-Instruct | FunSearch* | 3.02% | $3.07\%_{0.0008}$ |
| | QUBE (Ours) | **1.79%** | $\mathbf{2.76\%}_{0.0016}$ |
| Deepseek-coder-6.7b | FunSearch* | 3.09% | $3.19\%_{0.0011}$ |
| | QUBE (Ours) | 2.69% | $2.89\%_{0.0017}$ |

Table 3: Results of different LLMs on online bin packing OR3. Our method consistently outperforms FunSearch across different LLM variants.

We evaluate the following variants of our method:

**Parent Selection Only**: This variant adopts QUBE's parent selection strategy, selecting clusters with the top-2 $\tilde{Q}_t(\mathbb{C})$ at each timestep. The island reset mechanism is identical with FunSearch*.

**Quality Only**: This variant selects parents based on the top-2 $Q_t(\mathbb{C})$, which reflects our definition of sample quality without incorporating uncertainty. Island resets follow the FunSearch*.

**FunSearch***: Our replica of FunSearch as introduced in Section 4.3.

Table 2 reports both the best and average excess ratios (with standard deviation) across 10 runs for each variant. The performance gap between **FunSearch*** and **Quality Only** highlights the importance of using $Q_t(\mathbb{C})$ over raw scores $s(\mathbb{C})$ to evaluate sample quality. Unlike $s(\mathbb{C})$, which may be intuitively straightforward, $Q_t(\mathbb{C})$ provides an unbiased estimate of expected offspring quality, leading to more effective exploitation.

Further improvements observed in **Parent Selection Only** demonstrate the value of incorporating uncertainty into the priority criterion. By smartly allowing underrepresented clusters to be selected as parents, QUBE explores less visited regions of the function space that may yield high-quality heuristics over time. This mechanism enables adaptive exploration without sacrificing exploitation.

Finally, the performance difference between **QUBE** and **Parent Selection Only** underscores the impact of our island reset strategy. Unlike FunSearch, which resets islands based on current scores, QUBE targets islands unlikely to produce high-quality offspring. This forward-looking approach preserves exploratory potential and avoids prematurely discarding promising regions.

Together, these results validate QUBE's effectiveness in balancing exploitation and exploration, thus leading to robust and superior performance.

### 4.7 CHOICE OF LLMs

To assess the robustness of QUBE against variations in unrelated factors such as the choice of LLM, we conduct experiments on the OR3 dataset from OBP task. In addition to OpenCoder-8b-Instruct (Huang et al., 2024), we test with Deepseek-coder-6.7b (Guo et al., 2024), a smaller model with potentially weaker code generation capabilities.

As shown in Table 3, QUBE consistently outperforms FunSearch* regardless of the LLM used. While OpenCoder yields stronger results overall, the relative advantage of QUBE remains stable across models. These findings suggest that QUBE's performance gains stem from its algorithmic design rather than reliance on specific LLM capabilities, and that further improvements may be possible with more powerful models on complex tasks such as Cap Set.

### 5 CONCLUSION

In this paper, we analyze the limitations of FunSearch, a representative LLM+EA framework for heuristic optimization, and identify its shortcomings in balancing exploitation and exploration. To address these issues, we propose QUBE, a novel method inspired by the principles of UCB. QUBE introduces a quality-uncertainty trade-off mechanism that guides evolutionary search more effectively. Extensive experiments across multiple NP-complete problems demonstrate that QUBE consistently outperforms baseline methods. These results highlight QUBE's robustness and its potential to unlock stronger heuristic discovery, paving the way for broader applications of LLM-driven evolution in complex problem domains.

## 6 ETHICS STATEMENT

This research adheres to the ICLR Code of Ethics. Our work focuses on improving heuristic evolution methods through algorithmic design, without involving human subjects, sensitive personal data, or proprietary datasets. All experiments are conducted on publicly available benchmark datasets (e.g., OR-Library for Online Bin Packing, standard TSP instances, and Cap Set formulations), ensuring reproducibility and transparency.

The proposed method, QUBE, is designed to enhance the efficiency and robustness of heuristic discovery in combinatorial optimization. It does not introduce or amplify risks related to fairness, discrimination, privacy, or security. While our approach leverages large language models (LLMs), we do not rely on or modify their internal parameters, nor do we generate or deploy human-facing content. All model usage complies with licensing terms and responsible AI practices.

QUBE as well as FunSearch, requires generating codes using LLMs and running these codes on some test instances. This might have safety issues, since the code generated by LLM may have unpredictable outcome. In our experiment, we witnessed codes generated by LLM trying to modify (write and read) irrelevant local files. We tried our best to overcome this risk in our experiments by restricting permission to access local disk, running codes in safe namespaces, etc.

We acknowledge that improvements in automated heuristic generation may influence decision-making systems in broader applications. We encourage future work to consider domain-specific ethical implications when deploying such methods in sensitive contexts.

## 7 REPRODUCIBILITY STATEMENT

We have made extensive efforts to ensure the reproducibility of our results. The main text provides a detailed description of the QUBE framework (see Sections 3 and 4). All experimental settings, including datasets, evaluation metrics, and baseline comparisons, are described in Section 4 and further elaborated in Appendix D.1 and G. Hyperparameter settings and search procedures are reported in Appendix F and Table 4. Upon acceptance, we will release our source code and scripts as supplementary material to facilitate replication of our results. Any additional clarifications or instructions for running the experiments will be included in the code repository.

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

# A RELATED WORK

## A.1 HEURISTICS FOR MATH PROBLEMS

Heuristics are typically used to search solutions for NP-hard problems such as the Traveling Sales-man Problem (TSP) (Liu et al., 2023), online bin packing (OBP) (Coffman Jr et al., 1984), cap set problem (Grochow, 2019; Tao & Vu, 2006) etc. They guide the search direction to find rel-atively good solutions within a reasonable time. While it's hard to hand craft a good heuristic, hyper-heuristics algorithms (Burke et al., 2003) like EA can automatically optimize heuristics from a trivial on (Jia et al., 2023; Mei et al., 2023). Since the boost of deep learning, various relevant methods have been used to assist EA (Bengio et al., 2021; Hudson et al., 2022; Hottung et al., 2020).

## A.2 LLM+EA

The effectiveness of EA heavily relies on the ability of variation operators to generate diverse and promising new candidates, a process that typically demands substantial domain-specific knowl-edge (O'Neill et al., 2010). Recent research has explored the integration of EAs with LLM's gen-erative potential, termed LLM+EA methods (Lehman et al., 2024). These methods leverage the few-shot generation capabilities of LLMs as variation operators, extending their applications to di-verse domains such as neural architecture search (Chen et al., 2024), text-based tasks (Meyerson et al., 2023), optimization (Brahmachary et al., 2024), and molecular design (Wang et al., 2024).

Subsequent studies have focused on refining LLM+EA methodologies by enhancing LLM through prompting, reflection and other generation strategies. For instance, EoH (Liu et al., 2024) intro-duces five distinct prompts tailored for exploration and modification, moving beyond the single fixed prompt used in earlier approaches. Additionally, EoH suggests that LLMs should first gener-ate a textual description before implementing code. Similarly, ReEvo (Ye et al., 2024) incorporates LLM reflection into the process, enabling the model to generate improved samples based on insights derived from historical data. Despite these advancements, existing LLM+EA methods still face challenges in scalability, efficiency, and their applicability to more complex problems. Recently, Evotune (Šurina et al., 2025) proposed to finetune the LLM sampler using sample pairs collected during the sampling process with the DPO (Rafailov et al., 2023) objective. Despite their varying degrees of effectiveness, these methods have yet to address the fundamental issues inherent in the EA framework.

Concurrent to our work, Sankaran & McConky (2024) proposed UBS, which also incorporates UCB. However, their approach applies UCB solely for parent selection within a conventional EA frame-work. Moreover, their experiments are conducted on relatively small-scale problems (fewer than 200 iterations), whereas our study addresses challenging combinatorial optimization tasks that re-quire up to two million samples for effective resolution. Distinctively, we define sample quality based on its evolutionary advantage: specifically, the mean performance of its generated offspring. This diverges from all existing methodologies. To the best of our knowledge, our method QUBE, is the first to employ an online confidence-bound-driven selection mechanism grounded in a sample's evolutionary priority within this context.

## A.3 FUNSEARCH AND BEYOND

Existing LLM+EA methods have predominantly operated on a limited scale, typically generating fewer than 10,000 samples throughout the evolutionary process. These approaches have not yet fully leveraged the generative potential of LLMs or the evolution power of EAs. As a result, their applications have largely been confined to conventional combinatorial optimization problems, such as the TSP and OBP, which require relatively few evolutionary steps to yield meaningful results.

In contrast, FunSearch (Romera-Paredes et al., 2024) represents a significant leap in scaling LLM+EA methods, generating approximately 2.5 million samples during its evolutionary process. FunSearch extends beyond theoretical and mathematical domains, addressing complex and signif-icant challenges such as the cap set and admissible set problems. By significantly scaling up the generation of sample, FunSearch has demonstrated that LLM+EA algorithms can achieve state-of-

the-art (SOTA) solutions to exceptionally difficult problems, surpassing the capabilities of all prior LLM+EA methods.

## B    LIMITATIONS

Despite making non-trivial improvements on combinatorial optimization problems like online bin packing and TSP, our method fails to outperform heuristics searched by FunSearch (Romera-Paredes et al., 2024) on the cap set problems. Although this may potentially diminish the superiority of our method on large-scale complex problems, we have made every effort to demonstrate the advantage of our method over "FunSearch*" on the cap set problem under comparable settings. The performance of the best heuristics discovered is related to the choice of LLM, the number of samples generated and some random factors. Besides, to the best of our knowledge, no research work has ever surpassed or even tested the result of FunSearch (Romera-Paredes et al., 2024) in the cap set problem due to its extremely high computation requirements. We see this as an opportunity to further extend the capability and efficiency of LLM+EA methods.

Moreover, our method as well as FunSearch, requires generating codes using LLMs and running these codes on some devices. This might be dangerous, since the code generated by LLM may be unpredictable and hard to explain. In our experiment, we observed codes generated by LLM trying to modify (write and read) local files. We tried our best to overcome this risk in our experiments by restricting permission to access local disk, running codes in safe namespaces, etc.

## C    LLM USAGE STATEMENT

In this project, LLMs were used solely to assist with writing and polishing the manuscript. Specifically, LLMs were employed to improve clarity, grammar, and academic tone during the drafting and revision of textual content. No LLMs were used for research ideation, experimental design, data analysis, or generation of scientific content. All conceptual contributions, methodological innovations, and experimental results are entirely the work of the authors.

The authors take full responsibility for the content of this paper, including any text that may have been refined with the help of LLMs. No LLM qualifies for authorship under ICLR's Code of Ethics.

## D    MORE EXPERIMENT DETAILS

### D.1    CONSTRUCTION OF DATA

We list further details of our experiments here.

For OR datasets of online bin packing, we directly run our method and baseline methods on the test instances of each subset (OR1 $\sim$ OR4). The offline lower bound for each instance in these datasets is available, and the excess ratio for each subset is calculated directly using the sum of all used bins and the sum of all lower bounds of all instances.

For Weibull datasets of online bin packing, we generate 5 test instances for each setting following settings in Romera-Paredes et al. (2024), with 1k, 5k, 10k items each for Weibull1k, Weibull5k, Weibull10k respectively. Each bin's capacity is set to 100. The size of each item is sampled from Weibull(45, 3) distribution, clipped to 0$\sim$100, and finally rounded to an integer between 1 and 100. The offline lower bound for each instance in Weibull datasets is calculated following Martello & Toth (1990).

The input for the cap set problem is simply the number of dimensions $n$. Since the cap set problem is already solved for $n \leq 6$, we experimented with $n = 8$. Our method generates a heuristic within a guided greedy construction of cap set. Each heuristic can be evaluated through the size of the cap set found using itself.

The test instances for TSP are generated following the same setting as previous works (Kool et al., 2018; Liu et al., 2024). For each setting (TSP20, TSP50, TSP100) 1000 test instances are generated, each with 20, 50, or 100 locations randomly initialized from $[0, 1]^2$, respectively.

| | Hyperparameter | OBP | | Cap Set | TSP |
|---|---|---|---|---|---|
| | | OR | Weibull | | |
| LLM Samplers | Number of samplers | 16 | 16 | 16 | 16 |
| | LLM nucleus sampling $p$ | 0.95 | 0.95 | 0.95 | 0.95 |
| | LLM sampling temperature $t$ | 1.0 | 1.0 | 1.0 | 1.0 |
| | Samples generated per prompt: $n_s$ | 4 | 4 | 4 | 1 |
| | Total number of samples | 80K | 80K | 2M | 2K |
| Evaluators | Number of evaluators | 50 | 50 | 50 | 50 |
| | Timeout limit (in seconds) | 30 | 60 | 90 | 90 |
| DataBase | Number of islands: $n$ | 10 | 10 | 10 | 1 |
| | UIQ hyperparameter for uncertainty: $k$ | 0.0008 | 0.0001 | 32.0 | $10^{-5}$ |
| | Island reset interval: $T_{reset}$ | 32,768 | 32,768 | 262,144 | - |
| | Temperature for choosing sample: $T_{prog}$ | 1.0 | 1.0 | 1.0 | 1.0 |

Table 4: Implementation details for our method as well as baseline methods.

## D.2 HYPERPARAMETER SETTING

Apart from implementation details mentioned in Section 4.1, we list the hyperparameter settings in Table 4. One hyperparameter, specifically $k$ used in Equation 2 for UIQ, is searched for the optimal value. We show the results in Appendix F. The values of other hyperparameters are either identical to FunSearch Romera-Paredes et al. (2024) or carefully chosen to ensure the results are suitable for our implementation and hardware while also comparable among baselines.

## E MORE RESULTS FOR FIGURE 1

In Figure 1 of Section 1, we only show experiment results on online bin packing. We plot more experiment results in Figure 6. Our method finds a larger cap set than "FunSearch*" and outperforms all baseline methods on TSP100. Since the result on TSP20 and TSP50 is all 0 for all method, which is equal to the theoretical best, we are not showing them in plots.

## F HYPERPARAMETER SEARCH RESULTS

The value for the hyperparameters used in our method, namely UIQ's hyperparameter $k$, is searched. To search for the best value for $k$, we run experiments on "Parent Selection Only" method as described in Section 4.6. Apart from the cap set problem, each setting is run 10 times to calculate the average performance.

For OR dataset of OBP, we investigated that the appropriate value for $k$ should be between 0.01 to 0.0001 so as to balance the quality term and uncertainty term well. Experiments are run on OR3 dataset. We provide experiment results for $k$ in Table 5.

| $k$ | Best Run | Avg |
|---|---|---|
| 0.01 | 2.87% | 2.97% |
| 0.008 | 2.84% | 3.05% |
| 0.004 | 2.97% | 3.03% |
| 0.002 | 2.89% | 3.12% |
| 0.001 | 2.74% | 2.86% |
| 0.0008 | **2.59%** | **2.79%** |
| 0.0004 | 2.72% | 2.84% |
| 0.0002 | 2.68% | 2.82% |
| 0.0001 | 2.70% | 2.89% |

Table 5: Hyperparameter search result for $k$ on OR3 online bin packing. The optimal $k$ is 0.0008.

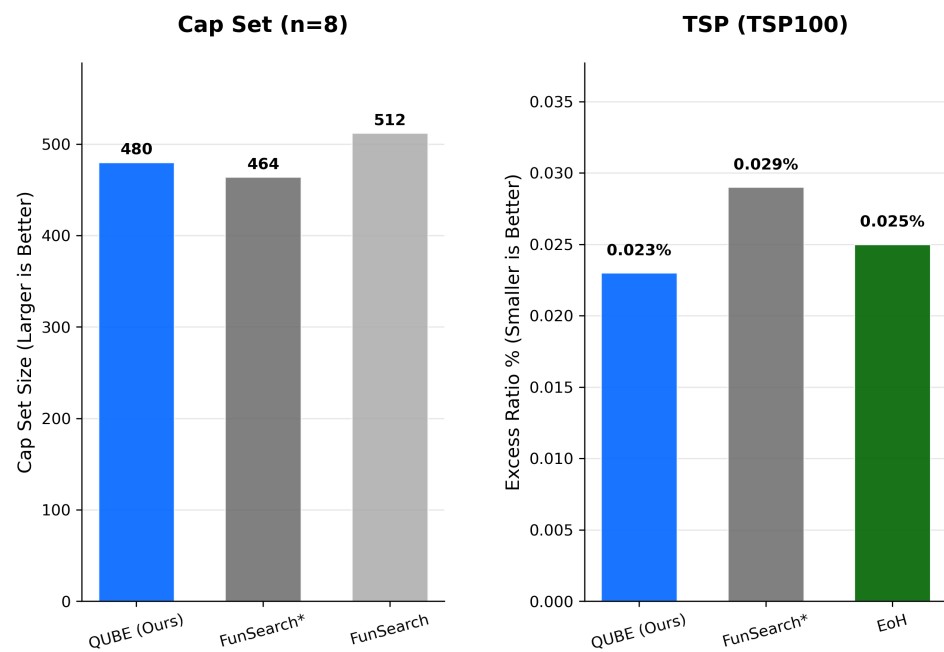

Figure 6: More experiment results on cap set n=8 and TSP100. For TSP a smaller excess ratio is better, while for cap set a larger found set size is better.Our method still shows superiority over baseline methods.

For Weibull dataset of OBP, we investigated that the appropriate value for $k$ should be between 0.001 to 0.00001 so as to balance the quality term and uncertainty term well. Experiments are run on Weibull5k dataset. We provide experiment results for $k$ in Table 6.

Similarly, for cap set problem, we experimented $kr$ within the range of 16 to 64. Since it cost heavily to run cap set experiments, we only run 5 runs for each setting and show the results in Table **??**.

## G    CODE SPECIFICATION FOR EACH TASK

In this section, we show the code specifications for each task. The function decorated with "@evolution" is evolved in experiments and the score of each function can be acquired by running the function decorated with "@run" on each test instance.

For online bin packing, the code specification is shown in Table 8. For the cap set problem the code specification is shown in Table 9. For TSP, the code specification is shown in Table 10.

| $k$ | Best Run | Avg |
|---|---|---|
| 0.001 | 1.73% | 1.86% |
| 0.0008 | 1.65% | 1.90% |
| 0.0004 | 1.67% | 1.83% |
| 0.0002 | 1.62% | 1.75% |
| 0.0001 | **1.54%** | **1.72%** |
| 0.00008 | 1.59% | 1.79% |
| 0.00004 | 1.64% | 1.82% |
| 0.00002 | 1.60% | 1.78% |
| 0.00001 | 1.70% | 1.88% |

Table 6: Hyperparameter search result for $k$ on Weibull5k online bin packing. The optimal $k$ is 0.0001.

| $k$ | Best Run | Avg |
|---|---|---|
| 16 | 464 | 452.8 |
| 32 | **464** | **464** |
| 48 | 464 | 451.2 |
| 64 | 448 | 448 |

Table 7: Hyperparameter search result for $k$ on cap set n=8. We use "Parent Selection Only" for experiment. The optimal value is 32.

## H  BEST HEURISTICS DISCOVERED

We show the best heuristics discovered by our method for each task here. The whole part of the function LLM samplers outputs are shown without any modification, which is why some part of the answers might sound nonsense.

For online bin packing OR1 the best heuristic discovered is shown in Table 11. For OR2, the best heuristic is shown in Table 12. For OR3, the best heuristic is shown in Table 13. For OR4, the best heuristic is shown in Table 14.

For cap set n=8, our best heuristic finds a cap set of 480 vectors. The corresponding heuristic is shown in Table 15.

## I  LLM PROMPTS

We write task-specific natural instructions for LLM samplers in MarkDown style, since the LLM we choose is capable of understanding and generating in MarkDown style. In all prompts shown below, "{Parent1}" and "{Parent2}" are replaced with two parents selected at each time step.

For online bin packing, the prompt we use is shown in Table 16. For cap set problem, the prompt we use is shown in Table 17. For TSP, the prompt we use is shown in Table 18.

```python
import os
import numpy as np

class BinPackProblem:
  def __init__(self, id, capacity, n_items, best_answer, items):
    self.id = id
    self.capacity = capacity
    self.n_items = n_items
    self.best_answer = best_answer
    self.items = np.array(items)
    assert len(items) == n_items
    bins = [capacity] * n_items
    self.bins = np.array(bins)

def get_valid_bin_indices(item, bins: np.ndarray) -> np.ndarray:
  return np.nonzero((bins - item) >= 0)[0]

def online_binpack(items: tuple[float, ...], bins: np.ndarray) -> tuple[
                                   list[list[float, ...], ...], np.
                                   ndarray]:
  packing = [[] for _ in bins]
  for item in items:
    valid_bin_indices = get_valid_bin_indices(item, bins)
    priorities = priority(item, bins[valid_bin_indices])
    best_bin = valid_bin_indices[np.argmax(priorities)]
    bins[best_bin] -= item
    packing[best_bin].append(item)
  packing = [bin_items for bin_items in packing if bin_items]
  return packing, bins

@run
def evaluate_binpack(problem):
  items = problem.items
  bins = problem.bins
  best_answer = problem.best_answer
  capacity = problem.capacity
  _, bins_packed = online_binpack(items, bins)
  solved_answer = (bins_packed != capacity).sum()
  cnt = best_answer - solved_answer
  ratio = cnt / best_answer
  return ratio

@evolution
def priority(item: float, bins: np.ndarray) -> np.ndarray:
  # Returns the priority with which we want to add 'item' to the bins
  return 0.0
```

Table 8: Code specification for online bin packing.

```python
"""Finds large cap sets."""
import itertools
import numpy as np

def solve(n: int) -> np.ndarray:
  """Returns a large cap set in `n` dimensions."""
  all_vectors = np.array(list(itertools.product((0, 1, 2), repeat=n)),
                                   dtype=np.int32)
  # Powers in decreasing order for compatibility with `itertools.product
                                   `, so
  # that the relationship `i = all_vectors[i] @ powers` holds for all `i
                                   `.
  powers = 3 ** np.arange(n - 1, -1, -1)
  # Precompute all priorities.
  priorities = np.array([priority(tuple(vector), n) for vector in
                                   all_vectors])
  # Build `capset` greedily, using priorities for prioritization.
  capset = np.empty(shape=(0, n), dtype=np.int32)
  while np.any(priorities != -np.inf):
    # Add a vector with maximum priority to `capset`, and set priorities
                                   of
    # invalidated vectors to `-inf`, so that they never get selected.
    max_index = np.argmax(priorities)
    vector = all_vectors[None, max_index]  # [1, n]
    blocking = np.einsum('cn,n->c', (- capset - vector) % 3, powers)  # [
                                   C]
    priorities[blocking] = -np.inf
    priorities[max_index] = -np.inf
    capset = np.concatenate([capset, vector], axis=0)

  return capset

@run
def evaluate(n: int) -> int:
  """Returns the size of an `n`-dimensional cap set."""
  capset = solve(n)
  return len(capset)

@evolution
def priority(element: tuple[int, ...], n: int) -> float:
  """Returns the priority with which we want to add `element` to the cap
                                   set."""
  return 0.0
```

Table 9: Code specification for cap set problem.

```
1026   import numpy as np
1027   import random
1028   import math
1029
1030   def euclidean_distance(city1, city2):
1031       return math.sqrt((city1[0] - city2[0])**2 + (city1[1] - city2[1])**2)
1032
1033   def calculate_total_distance(route, distance_matrix):
1034       return sum(distance_matrix[route[i]][route[i+1]] for i in range(len(
1035                                       route)-1)) + distance_matrix[
1036                                       route[-1]][route[0]]
1037
1038   def two_opt(route, distance_matrix):
1039       best_route = route.copy()
1040       improved = True
1041       while improved:
1042           improved = False
1043           for i in range(1, len(route)-2):
1044               for j in range(i+1, len(route)):
1045                   if j-i == 1: continue
1046                   new_route = route[:i] + route[i:j][::-1] + route[j:]
1047                   if calculate_total_distance(new_route, distance_matrix) <

                                                   calculate_total_distance
                                                   (best_route,
                                                   distance_matrix):
1048                       best_route = new_route
1049                       improved = True
1050           route = best_route
1051       return best_route
1052
1053   @run
       def guided_local_search(cities, max_iterations=100, alpha=0.1):
1054       num_cities = len(cities)
1055       distance_matrix = np.zeros((num_cities, num_cities))
1056       for i in range(num_cities):
1057           for j in range(i+1, num_cities):
1058               distance_matrix[i][j] = distance_matrix[j][i] =
                                               euclidean_distance(
                                               cities[i], cities[j])
1060       init_distance_matrix=copy.deepcopy(distance_matrix)
1061       # Initialize route
1062       route = list(range(num_cities))
1063       best_route=route
1064       # Initialize penalties
1065       penalties = np.zeros((num_cities, num_cities))
       for iteration in range(max_iterations):
1066           # Local search with 2-opt
1067           route = two_opt(route, distance_matrix)
1068           # Update route
           if calculate_total_distance(route, init_distance_matrix) <
1069                                               calculate_total_distance(
1070                                               best_route,
1071                                               init_distance_matrix):
               best_route=route
1072           # Evolve distance_matrix
1073           distance_matrix=distance_matrix+update_dist(distance_matrix,
1074                                               best_route)
1075       return best_route, calculate_total_distance(best_route,
1076                                               init_distance_matrix)
1077
1078   @evolution
       def update_dist(distance_matrix, current_route):
1079       ''' calculates an update to current distance matrix. '''
       return np.zeros_like(distance_matrix)
```

Table 10: Code specification for TSP.

```python
def priority(item: float, bins: np.ndarray) -> np.ndarray:
  penalty_factor_v3 = 0.7

  D_item_val, C_int_fit, B_valid_region, a_of_K2_val = 4.5, 3.5, 2.6, 4.7

  item_weight = item / 4650

  scores = np.zeros(len(bins))

  K_values = np.array([0.28, 0.31, 0.35])

  B_values = np.array([0.15, 0.3, 0.25])

  b_weights = np.array([2750/4650, 2950/4650, 3050/4650, 3150/4650])

  for index, bin_num in enumerate(bins):
    quantity_1D = index * bin_num
    calc_2D_quantity = bin_num * bin_num

    if index <= 3400:
      b_weight = b_weights[0]
    elif index<=3800:
      b_weight = b_weights[1]
    else:
      b_weight = b_weights[3]

    P_item = (index * b_weight) * (quantity_1D / calc_2D_quantity)

    # Further improvements here.

    improved_P_item = P_item * (index**52) * (item_weight**67) * (index**
                                2.5) * (item_weight**4.0) * (
                                index**3.4) * (item_weight**3.2)
                                * (index**3.0) * (item_weight**
                                3.3)

    valid_region = abs(quantity_1D / calc_2D_quantity - 1)

    if index <= 3000:
      K = (K_values[0] * penalty_factor_v3) + ((1 - penalty_factor_v3) *
                                K_values[1])
    elif index<=3800:
      K = K_values[1]
    else:
      K = K_values[2]

    if index <= 3500:
      B_val = (B_values[0] * penalty_factor_v3) + ((1 - penalty_factor_v3
                                ) * B_values[1])
    elif index<=3800:
      B_val = B_values[1]
    else:
      B_val = B_values[2]

    intersection_fit = ((index * item_weight / (abs(bin_num - item)))**42
                                ) * K * 2400000

    improved_D_item_val = D_item_val * ((bins[index]/item) ** 2.8) * (1.0
                                + index / 95000)
    improved_C_int_fit = C_int_fit * (95 / (index+6))
    improved_B_valid_region = B_val + (1-B_val) * (valid_region**2.5)
    improved_a_of_K2_val = a_of_K2_val / (1 + index / 95000)

    P_final = improved_D_item_val * ((improved_P_item + C_int_fit *
                                intersection_fit) / (
                                improved_B_valid_region * (
                                improved_a_of_K2_val +
                                valid_region)))

    scores[index] = P_final
```

```python
def priority(item: float, bins: np.ndarray) -> np.ndarray:
  bins_difference = np.abs(bins - item)

  low_threshold, high_threshold = 8, 23
  diff_mid = (high_threshold + low_threshold) / 2

  p_vect4 = np.where(bins_difference <= low_threshold, bins_difference *
                                        (-1) * 22,
                  np.where(bins_difference <= diff_mid, bins_difference *
                                              (-1) * 34,
                  np.where(bins_difference <= high_threshold,
                                              bins_difference *
                                              (-1) * 46,
                                              bins_difference *
                                              (-1) * 2)))

  p_vect4[np.abs(bins_difference) <= high_threshold / 2] += 35
  p_vect4[np.abs(bins_difference) <= diff_mid] += 50
  p_vect4[np.abs(bins_difference) <= low_threshold + high_threshold / 2]
                              += 64

  for i, val in enumerate(bins_difference):
    if val <= 25:
        bins_difference[i] = bins_difference[i] * (i + 1) * 72
    else:
        break

  if np.any(np.abs(np.where(bins_difference <= 25, bins_difference * (-1)
                                        * 100, bins_difference * (-1) *
                                        13)) <= 150):
    p_vect4[np.abs(np.where(bins_difference <= 25, bins_difference * (-1)
                                        * 95, bins_difference * (-1) *
                                        13)) <= 150] += 42

    best_global = sorted(p_vect4)
    best_three_values = best_global[0:3]
    worst_bin_index = np.where(p_vect4 == max(best_three_values))[0][0]

    if worst_bin_index < len(p_vect4):
      p_vect4[worst_bin_index] = min(p_vect4) * 0.98

  return p_vect4
```

Table 12: The best heuristic searched by our method for OR2 online bin packing.

```
1188  def priority(item: float, bins: np.ndarray) -> np.ndarray:
1189    probabilities = np.zeros(len(bins), dtype=float)
1190
1191    for i in range(len(bins)):
1192      current_bin_space = bins[i]
1193
1194      if item <= current_bin_space:
1195        remainingSpaceFactor = current_bin_space / (current_bin_space +
                                                        item)
1196        enhanced_load_factor = item/current_bin_space
1197
1198        # Improved estimation formula: f(x) = a * x ** p * exp(x)
1199
              """
1200          Non-uniform impact approach based on the load intensity:
1201          Enhance the evaluated importance of loading by approaching loader-
1202                                            bins outcomes.
1203          """
              additional_impact_factor = 0.00
1204
1205        if enhanced_load_factor < 0.95:
1206          modified_priority = (0.99 * ((remainingSpaceFactor / (1 -
1207                                          enhanced_load_factor)) - 2.
1208                                          55 +
1209                                          additional_impact_factor) *
1210                                          1500 - 95 / (
1211                                          remainingSpaceFactor ** 1.25
                                            )) * (130 + 0.0095 * i) * np
1212                                          .exp(-i * 0.022)
1213        elif enhanced_load_factor < 0.99:
1214          modified_priority = (1.00 * ((remainingSpaceFactor / (1 -
1215                                          enhanced_load_factor)) - 2.
1216                                          45 +
1217                                          additional_impact_factor) *
1218                                          1600 - 45 / (
1219                                          remainingSpaceFactor ** 1.30
                                            )) * (140 + 0.0105 * i) * np
1220                                          .exp(-i * 0.022)
1221        else:
              modified_priority = (1.01 * ((remainingSpaceFactor / (1 -
1222                                          enhanced_load_factor)) - 2.
1223                                          35 +
1224                                          additional_impact_factor) *
1225                                          1700 - 35 / (
1226                                          remainingSpaceFactor ** 1.35
                                            )) * (160 + 0.0115 * i) * np
1227                                          .exp(-i * 0.023)
1228
1229        # Added/displaced non-uniform interpolated/smooth kernel-duty
1230                                            system aspects
1231        modified_priority -= 500 + 70 * np.cos(enhanced_load_factor + 0.07)
1232                                        + 600 * np.tanh(2.84 * (
1233                                        enhanced_load_factor - 0.93))
1234                                        + 80 * np.cos(2 * i / len(bins
                                          )) + 880 * np.sin(2 * i / len(
1235                                        bins))
1236
1237        # Adjust differently for injected non-trivial items using maximum
1238                                            performance complexity system
1239
1240        modified_priority -= 35 * (1-enhanced_load_factor) ** 0.98
1241
          # Insert updated, optimized weights for different scenarios

          probabilities[i] = modified_priority

    return probabilities                23
```

Table 13: The best heuristic searched by our method for OR3 online bin packing.

```python
def priority(item: float, bins: np.ndarray) -> np.ndarray:
  def improved_prior_func(_value):
    if _value < item / 9:
      if bins.size > 700:
        return 260**(35 * item / 350 - 2.5 * _value)
      elif bins.size > 350:
        return 140**(30 * item / 350 - 1 * _value)
      else:
        return 140**(50 * item / 350 - 2.5 * _value)    # Colocalization

    elif _value < item / 5:
      if bins.size > 700:
        return 180**(35 * item / 350 - 1 * _value)
      elif bins.size > 350:
        return 110**(40 * item / 350 - 0.5 * _value)   #Quorum sensing
      else:
        return 140**(40 * item / 350 - 0.6 * _value)    # Quorum sound
                                              BiellLIF

    elif _value < item:
      if bins.size > 700:
        return 95 * item /(145 + item)
      elif bins.size > 350:
        return 80 * item /(125 + item)
      else:
        return 80 * item /(130 + item)      #Rotulina colleague
                                            asymmetrically restructuring
                                             translators replication
                                            achieved in cell-process

    else:
      if bins.size > 700:
        return 105 * item /(130 + item)
      elif bins.size > 350:
        return 95 * item /(120 + item)
      else:
        return 95 * item /(110 + item)        #Biulation sncRNA
                                              oscillations

  return np.vectorize(improved_prior_func)(bins - item)
```

Table 14: The best heuristic searched by our method for OR4 online bin packing.

```
def solve(n: int) -> np.ndarray:
  score = np.sum(element) * 220.00 * 3.0
  zeros = [idx for idx, val in enumerate(element) if val == 0]
  # If there are at least two zeros.
  if len(zeros) >= 2:
    score = np.abs(np.sum(zeros)) * 230.00 * 2400.0
  # If there are at least three zeros.
  if len(zeros) >= 3:
    d = np.array(zeros)[1:] - np.array(zeros)[:-1]
    d_sorted = np.sort(d)
    r = d_sorted[-1]
    if r % 2 == 0:
      score = np.abs(zeros[0] - zeros[1]) * 250.00 * 3400.0
  # If there are at least four zeros.
  if len(zeros) >= 4:
    score = np.sum(element) * 260.50 * 35.0
  # If there are more than three zeros and less than six zeros.
  if len(zeros) > 3 and len(zeros) < 6:
    score += 35000.0 * np.sum(zeros)
  # If there are more than five zeros and less than nine zeros.
  if len(zeros) > 5 and len(zeros) < 9:
    score += 36000.0 * np.sum(element)
  # If there are six or more zeros.
  if len(zeros) >= 6:
    score *= np.sum(np.array(element))
  # Add some score based on the minimum and maximum elements.
  score += np.sum(element) * np.min(np.array(element[:2])) * np.max(np.
                                    array(element)) * 100.00
  # If there is one zero, multiply the score by 120.
  if len(zeros) == 1:
    score *= 120.0
  # Subtract some value based on the sum of the elements.
  score -= np.sum(element) * np.sum(element[:2]) / 4.5
  # If there are no zeros, multiply the score by 115.
  if len(zeros) == 0:
    score *= 1.15
  # Multiply the score by 40.
  score *= 40.00
  # If there are seven or more zeros, add some value to the score.
  if len(zeros) >= 7:
    score += np.sum(element) * 250.00 * 120.0
    score *= 1.85
  if len(zeros) > 9 and len(zeros) < 12:
    score += np.sum(element) * 260.50 * 90.0
  # If there are twelve or more zeros, add some value to the score.
  if len(zeros) >= 12:
    score += np.sum(element) * 280.50 * 140.0
  if len(zeros) > 14:
    score *= np.sum(zeros)
  # Multiply the score by the maximum element plus 40.
  score *= np.max(np.array(element)) + 40.00
  if np.sum(element) <= 12:
    score *= 1.75
  # If there are five or fewer zeros, multiply the score by 27.
  if len(zeros) <= 5:
    score *= 27.0
  # Add 12000 to the score.
  score += 12000.0
  # If there are ten or fewer zeros, add 20000 to the score.
  if len(zeros) <= 10:
    score += 20000.0
  # If there are fifteen or fewer zeros, add 30000 to the score.
  if len(zeros) <= 15:
    score += 30000.0
  # Further improved version of 'priority_v2'.
  score *= 1.75
  # Final improvement of the score.
  score *= 1.45
  return score
```

25

Table 15: The heuristic searched by our method that leads to a cap set of size 480 on n=8

Online 1D bin packing problem is a combinatorial optimization problems. The goal of online bin packing is to assign each of a series of items into the smallest number of fixed-sized bins. Generally, heuristics are used to solve online bin packing efficiently. Priority function is defined in heuristic to help rank and search for best candidates.

You are given two priority functions "priority_v0" and "priority_v1", then you are asked to complete the following priority function "priority_v2" such that it is an improved version of "priority_v1". This priority function will be used in heuristic to ranks the priority of bins given incoming item.

Here are the requirements:
1. Just complete the "priority_v2" function and do note answer anything else.
2. Do not use "print" function in your answer.

```python
# Finds good assignment for online 1d bin packing.
import numpy as np

def priority_v0(item: float, bins: np.ndarray) -> np.ndarray:
    """ Returns the priority with which we want to add 'item' to the bins """
{Parent1}

def priority_v1(item: float, bins: np.ndarray) -> np.ndarray:
    """ Improved version of priority_v0 """
{Parent2}

def priority_v2(item: float, bins: np.ndarray) -> np.ndarray:
    """ Improved version of priority_v1 """
```

Table 16: Prompt Template for online bin packing

1404
1405
1406
1407
1408
1409
1410
1411
1412
1413
1414
1415
1416
1417
1418
1419
1420
1421
1422
1423
1424
1425
1426
1427
1428
1429
1430
1431
1432
1433
1434
1435
1436
1437
1438
1439
1440
1441
1442
1443
1444
1445
1446
1447
1448
1449
1450
1451
1452
1453
1454
1455
1456
1457

The cap set problem calculates the largest possible set of vectors in $\mathbb{Z}_3^n$ (known as a cap set) such that no three vectors sum to zero. Geometrically, no three points of a cap set lie on a line.

Generally, heuristics can be used to solve cap set problem. Priority function for solving the cap set problem ranks the priority with which we want to add a vector into the cap set.

Given two priority functions "priority_v0" and "priority_v1" where "priority_v1" is an improved version of "priority_v0", your task is to complete the following function priority_v2 such that it is an improved version of priority_v1. Just complete the code and do not answer anything else. Do not use any `print` function in your answer.

Here are the requiremnets:
1. Just complete the "priority_v2" function and do note answer anything else.
2. Do not use "print" function in your answer.

``` python
# Find large cap sets
import numpy as np
import itertools
def priority_v0(n: int) -> np.ndarray:
    """ Returns a large cap set in 'n' dimensions."""
{Parent1}

def priority_v1(n: int) -> np.ndarray:
    """ Improved version of priority_v0 """
{Parent2}

def priority_v2(n: int) -> np.ndarray:
    """ Improved version of priority_v1 """
```

Table 17: Prompt Template for cap set problem

TSP problem finds shortest paths that travels all places and return to the starting point. Guided local search can be used to iteratively update solution to TSP problems. A function updates the distance matrix according to current shortest paths, such that further local search on the updated distance matrix may lead to better answer.

You are given two update functions "update_dist_v0" and "update_dist_v1", then you are asked to complete the following priority function "update_dist_v2" such that it is an improved version of "update_dist_v1". This priority function will be used in heuristic to ranks the priority of bins given incoming item.

Here are the requirements:
1. Just complete the "update_dist_v2" function and do note answer anything else.
2. Do not use "print" function in your answer.

```python
import numpy as np
import random
import math
import copy

def update_dist_v0(distance_matrix ,current_route):
    """ Updates the distance matrix according to current best route searched"""
{Parent1}

def update_dist_v1(distance_matrix ,current_route):
    """ Improved version of update_dist_v0 """
{Parent2}

def update_dist_v2(distance_matrix ,current_route):
    """ Improved version of update_dist_v1 """
```

Table 18: Prompt Template for TSP.