# OpenReview forum: "QUBE: Enhancing Automatic Heuristic Design via Quality-Uncertainty Balanced Evolution"
_ICLR.cc/2026/Conference — ICLR 2026 Conference Withdrawn Submission_

### Official Review · Reviewer_AKSa · 2025-10-31

**Soundness:** 1
**Presentation:** 2
**Contribution:** 2
**Rating:** 2
**Confidence:** 4

**Summary:**

The paper proposes QUBE, an LLM+EA framework that replaces FunSearch’s score-only priority with a Quality-Uncertainty Trade-off Criterion (QUTC). QUTC uses Uncertainty-Inclusive Quality (UIQ), i.e. the average performance of a cluster’s offspring, plus a UCB exploration bonus to select parents and decide island resets. Diagnostics (RBS for exploitation, RPC for exploration) demonstrate more stable progress than FunSearch. QUBE demonstrates generally better results than EoH.

**Strengths:**

- The paper presents a simple but coherent idea that can be useful.

- LLMs used are local, which is appreciated

- Ablations studies and hyperparameter analyses are comprehensive.

**Weaknesses:**

1. Baselines are largely outdated. EoH was first released 1 and a half years ago and is the latest one compared against; e.g., there is no comparison to ReEvo (which is also cited) or more recent baselines.

2. Comparison with EoH is misleading. The authors report EoH results from the paper without re-running the code with their own LLM. For instance, EoH uses, to my knowledge, many fewer LLM calls and evaluations, which makes the comparison unfair

3. QUBE does not perform the original version of Funsearch in Cap Set, which is puzzling since the reproduced version was supposed to be generally better, but not in this case.

4. Comparison is done with the best out of 10 runs; reporting the average would be more fair.


Overall, with recent works in the literature and inadequate comparisons, I believe the paper is not yet ready for acceptance.

**Questions:**

1. What would be the performance against EoH with the same LLM and the same number of samples?

2. For the exploration proxy, I believe RPC (token edit distance to the nearest parent) is a weak proxy for functional novelty; large edits can be semantically trivial and vice versa. Can you comment on whether there could be better approaches?

3. Could you report some comparisons under the same number of samples against at least EoH using the same LLM?

---

### Official Review · Reviewer_HSQt · 2025-11-01

**Soundness:** 1
**Presentation:** 2
**Contribution:** 1
**Rating:** 2
**Confidence:** 4

**Summary:**

This paper introduces Quality-Uncertainty Balanced Evolution (QUBE), a novel method for automatically designing heuristics for NP-hard problems. The authors identify a key limitation in the state-of-the-art LLM+EA framework, FunSearch, arguing its priority criterion fails to effectively balance exploration and exploitation. To address this, QUBE proposes a new priority mechanism called the Quality-Uncertainty Trade-off Criterion (QUTC). QUTC evaluates potential parents (heuristics) not by their own performance, but by a UCB-inspired metric called Uncertainty-Inclusive Quality (UIQ), which considers both the historical performance of their offspring (evolutionary potential) and the uncertainty due to how frequently they have been sampled. Through extensive experiments on problems like Online Bin Packing, TSP, and Cap Set, the authors demonstrate that QUBE consistently outperforms FunSearch and other baselines, attributing this success to a more principled balance between exploiting promising search areas and exploring novel ones.

**Strengths:**

- The paper provides a clear and compelling analysis of FunSearch's limitations.
- The proposal for balancing exploration and exploitation in AHD is critical to improve the efficiency.

**Weaknesses:**

- The paper's primary empirical claim rests on outperforming FunSearch. While the authors commendably reproduce FunSearch (as FunSearch*) for a controlled comparison, this rigor is not extended to other baselines. The comparison to EoH, for instance, relies on results cited from the publication, which likely used a different LLM, computational budget, and experimental setup. To make a convincing case for state-of-the-art performance, the authors should have implemented or re-run other contemporary LLM+EA methods like EoH, ReEvo, HSEvo, MCTS-AHD under the exact same conditions (LLM, number of samples, hardware) as QUBE.

- The core mechanism of QUBE is an application of the UCB algorithm to guide parent selection. While effective, the paper does not sufficiently situate this contribution within the existing literature on UCB/MCTS for program or heuristic synthesis. The authors mention concurrent work (UBS) but fail to discuss or differentiate their method from potentially relevant prior work (e.g., MCTS-AHD) that has also employed confidence-bound-based exploration. This makes it difficult to assess the precise novelty of the contribution beyond a successful application of a known technique.

- Several key aspects of the methodology lack the detail required for full understanding and reproducibility: 1) The quality metric $Q_t(C)$ is undefined for a new cluster with no children. The paper does not specify how this crucial value is initialized, which would heavily influence whether a novel heuristic is ever selected as a parent. 2) The RPC metric, used to motivate the work, is vaguely defined. It is unclear how the "nearest parent" is selected from two candidates or which specific "token-level edit distance" algorithm is used, making the analysis in Figure 4 difficult to reproduce. 3) The process of grouping samples into clusters based on identical outputs is not detailed. As the number of clusters grows, the naive approach of comparing a new sample to all existing clusters becomes a performance bottleneck. The data structure and algorithm for this process should be explained.

- The hyperparameter k requires extensive, problem-specific tuning, with optimal values spanning several orders of magnitude (from 0.0001 to 32.0). This high sensitivity undermines the "automatic" nature of the method and presents a practical barrier to applying QUBE to new problems without a costly search.

- While QUBE outperforms its own reproduction (FunSearch*), it fails to surpass the original SOTA result from the FunSearch paper. Although the authors correctly cite the extreme computational cost as a barrier, the fact remains that on the most difficult benchmark presented, QUBE has not yet demonstrated clear superiority over the original state-of-the-art, which slightly tempers the paper's otherwise strong claims.

**Reference**

1. Ye, Haoran, et al. "Reevo: Large language models as hyper-heuristics with reflective evolution." Advances in neural information processing systems 37 (2024): 43571-43608.
1. Dat, Pham Vu Tuan, Long Doan, and Huynh Thi Thanh Binh. "Hsevo: Elevating automatic heuristic design with diversity-driven harmony search and genetic algorithm using llms." Proceedings of the AAAI Conference on Artificial Intelligence. Vol. 39. No. 25. 2025.
1. Zheng, Zhi, et al. "Monte Carlo Tree Search for Comprehensive Exploration in LLM-Based Automatic Heuristic Design." Forty-second International Conference on Machine Learning.

**Questions:**

- Regarding the island reset mechanism: the paper states that underperforming islands are reinitialized by sampling from a cluster of a randomly chosen surviving island. What is the motivation for choosing a random survivor instead of the globally best-performing island? Have you experimented with using the global best, and if so, how did it affect performance?
- Could you elaborate on the data structure and algorithm used to implement the clustering of samples based on identical outputs? How do you ensure this process remains efficient as the number of distinct clusters grows large?

---

### Official Review · Reviewer_GASa · 2025-11-02

**Soundness:** 3
**Presentation:** 2
**Contribution:** 2
**Rating:** 2
**Confidence:** 4

**Summary:**

The paper proposes QUBE, a method that addresses the limitations of FunSearch in balancing exploitation and exploration. QUBE employs a QUTC mechanism to select clusters, balancing quality and uncertainty, and achieves competitive results in three combinatorial optimization problems.

**Strengths:**

- The details of the proposed QUBE are clearly demonstrated.

- The effectiveness of QUBE is well illustrated.

**Weaknesses:**

1. The technical contribution of this paper is limited. The proposed CUBE method is essentially a variant of FunSearch, with the only distinction lying in the choice of cluster in each island.

2. The paper is not well structured and written that making it hard to follow up:
(i) Section 2.3 analyzes the proposed method with FunSearch before it is detailed in Sec 3.
(ii) Line 260-262, “We first introduce UIQ ... before the current timestep” -- I understand what these sentences mean, though they are hard to understand at first glance, as a new concept of “timestep” is used without being mentioned before.

3. The “function space” described in the paper is not accurate, as the proposed methods are actually optimizing heuristics in “language space” or “code space”.

4. Some phrases are used casually in the text:
(i) Line 77, “the priority criterion behind FunSearch...” -- I did not find any operation correlated with “priority criterion” in FunSearch’s paper.
(ii) Line 158, “structural novelty” -- On one hand, RPC does not consider the structural differences (e.g., the difference between two AST trees) between two code snippets; it measures the edit distance between tokens of code. On the other hand, the distance (or distinction) between tokens does not necessarily reflect their “novelty”.

5. The LLM used in this paper is outdated (DeepSeek-coder-6.7B, OpenCoder-8B-Inst), and incorporating recent LLMs is thereby expected.

6. The experiment section lacks comparison with state-of-the-art, e.g., AlphaEvolve, ReEvo, MCTS-AHD, etc.

**Questions:**

The author should address the comments in the Weaknesses section; a few minor issues are as follows.

Typo:
Line 117 “timeoutsare” -> “timeouts are”

---

### Official Review · Reviewer_pF7A · 2025-11-06

**Soundness:** 1
**Presentation:** 2
**Contribution:** 2
**Rating:** 2
**Confidence:** 4

**Summary:**

The paper proposes QUBE, an extension of FunSearch that introduces an Uncertainty-Inclusive Quality (UIQ) metric and a Quality–Uncertainty Trade-off Criterion (QUTC) to balance exploration and exploitation when evolving heuristics using large language models. The method is applied to Online Bin Packing, Travelling Salesman, and Cap Set problems, showing competitive or improved results over FunSearch and EoH. The paper also provides ablation studies and proxy analyses (RBS and RPC) to study search dynamics.

**Strengths:**

1. The motivation to balance exploration and exploitation is strong and well grounded.
2. The quantitative analysis of FunSearch’s limitations on exploration and exploitation balance is inspiring and valuable to the community.
3. The uncertainty bonus design based on the UCB principle is interesting and conceptually solid.

**Weaknesses:**

1. The description in Section 2.1 that “a randomly selected island undergoes evolution” appears inconsistent with the original FunSearch design, where all islands evolve synchronously and island replacement is score-driven rather than random. If QUBE adopts a stochastic sequential scheduling scheme, this constitutes a methodological deviation that should be clearly stated and ablated for fairness. Otherwise, the sentence may misrepresent baseline behavior.

2. The description at L151 that “RBS measures the highest score” conflicts with the rest of the paper’s metric orientation, where lower numerical values (e.g., excess ratio in Fig. 1) indicate better performance. Unless the authors internally negate or normalize scores (e.g., reward = -cost) before computing RBS/RPC, the interpretation of “high performance” as “high score” is inconsistent. This should be clarified in Sec. 2.3 with an explicit definition or sign convention.

3. I personally like the design of the uncertainty score in the UIQ. However, as the core technique contribution of this paper, I would suggest discussing the effect of the proposed function in more detail. For example, compare with simple linear increasing uncertainty, how good it is. In the meantime, the sensitivity analysis for this function would also be essential to show the effectiveness of this design.

4. $T_{\text{prog}}$ is fixed (0.001, Table 4) without justification or sensitivity analysis. As this parameter directly affects intra-cluster selection and heuristic expressiveness, the lack of ablation leaves its influence unclear.

5. I have a few concerns about the experiment's results. Table 1 reports 0 % gap for TSP20 and TSP50, which would imply exact optimality. The manuscript does not clarify whether this value is due to rounding, normalization (e.g., relative to the best baseline rather than the true optimum), or the simplicity of the dataset.

6. About baselines, the paper compares only to FunSearch and partially to EoH, omitting stronger recent baselines like ReEvo, GP-LLM, etc, and more recently, AHD works.

7. Even though authors say that the code will be available upon acceptance, there is no code reference for the paper review. This raises some concerns about the reproducibility of this paper. Also, some prompts in the appendix contain error codes, like Table 17, where, where there are “?” within the prompt.

[Minors]
- L249: “reset” should be in regular font, not italic, since it is not a variable. Same for L291.
- Tables should follow the ICLR official format.
- Some appendix tables are cropped (pages 21 and 25), which hurts readability.

Overall, I do like the exploration and exploitation discussion and related technique. But I think the current manuscript is not ready for the ICLR.

**Questions:**

1. In Figures 1 and 2, what is the meaning of the shaded regions behind the curves? For example, in Figure 1 (OR2), the FunSearch shadow line appears to have a consistent offset.

2. The RBS curves in Figure 1 fluctuate instead of monotonically increasing, though Sec. 2.3 defines RBS as “the highest score.” If RBS is computed within a finite window (e.g., last K iterations) rather than cumulatively, please state this explicitly. Otherwise, the oscillations might suggest a misimplementation or normalization artifact. Could you explain the reason behind this?

---

### Note · Authors · 2025-11-12

I have read and agree with the venue's withdrawal policy on behalf of myself and my co-authors.